# Is Explanation the Cure?
# Misinformation Mitigation in the Short Term and Long Term

**Yi-Li Hsu[1,2], Shih-Chieh Dai[3], Aiping Xiong[4], Lun-Wei Ku[1]**
[1]Institute of Information Science, Academia Sinica
[2]Department of Computer Science, National Tsing Hua University
[3]The University of Texas at Austin
[4]The Pennsylvania State University
yili.hsu@iis.sinica.edu.tw

## Abstract

With advancements in natural language processing (NLP) models, automatic explanation generation has been proposed to mitigate misinformation on social media platforms in addition to adding warning labels to identified fake news. While many researchers have focused on generating good explanations, how these explanations can really help humans combat fake news is under-explored. In this study, we compare the effectiveness of a warning label and the state-of-the-art counterfactual explanations generated by GPT-4 in debunking misinformation. In a two-wave, online human-subject study, participants (N = 215) were randomly assigned to a control group in which false contents are shown without any intervention, a warning tag group in which the false claims were labeled, or an explanation group in which the false contents were accompanied by GPT-4 generated explanations. Our results show that both interventions significantly decrease participants' self-reported belief in fake claims in an equivalent manner for the short-term and long-term. We discuss the implications of our findings and directions for future NLP-based misinformation debunking strategies.

## 1 Introduction

Misinformation (or fake news) refers to false statements or fabricated information that is spread on social media (Wu et al., 2019). The spread of misinformation has posed considerable threats to individuals and society, such as political elections (Allcott and Gentzkow, 2017; Grinberg et al., 2019; Lytvynenko and Craig, 2020), and public health (Swire-Thompson et al., 2020), especially during the COVID-19 pandemic (Roozenbeek et al., 2020; Loomba et al., 2021). Because of the negative impact, considering effort has been devoted to devising methods to mitigate misinformation, such as computation-based detection and prevention. Traditional fact-checking approaches are labor-intensive and time-consuming, often requiring domain expertise. Given advances in natural language processing (NLP), there is an ongoing shift towards NLP-based solutions such as fake news detection (Zhang and Ghorbani, 2020; Zhou and Zafarani, 2020; Shu et al., 2019) and generation of fact-checked, counterfactual explanations using natural language generation (NLG) models (Dai et al., 2022). Decision-aid methods have also been proposed and deployed to warn users when a piece of fake news has been identified. Warning labels (or tags) have been widely adopted by social media platforms such as Facebook[1], Twitter (X)[2], and TikTok[3] to mitigate humans' belief in misinformation.

Tag- and explanation-based methods are both effective in debunking fake news (Epstein et al., 2022; Moravec et al., 2020; Lutzke et al., 2019). However, few studies have compared tag-based and machine-generated explanations. Furthermore, studies on fake news debunking strategies have primarily focused on short-term effects instead of long-term effects (Epstein et al., 2022; Moravec et al., 2020; Lutzke et al., 2019). Thus we propose a comprehensive evaluation of both the short- and long-term effectiveness of these debunking strategies, specifically in the context of real-world news in the United States. For the explanation-based debunking strategy, we focus on counterfactual explanations, which have demonstrated greater effectiveness than summary-based explanations in mitigating the spread of misinformation (Dai et al., 2022). With the improvements of large language models (LLM) from OpenAI, GPT-3 (Brown et al.,

---

[1]https://www.facebook.com/journalismproject/programs/third-party-fact-checking/new-ratings
[2]https://blog.twitter.com/en_us/topics/company/2022/introducing-our-crisis-misinformation-policy
[3]https://www.tiktok.com/community-guidelines/en/integrity-authenticity/

2020) already generates explanations acceptable to humans (Wiegreffe et al., 2021). Here we employ the GPT-4 model, the latest state-of-the-art LLM from OpenAI, to generate all counterfactual explanations (OpenAI, 2023).

To investigate the effectiveness of debunking strategies, we focus on their impact over varied time frames. We aim to study the following research questions:

1. How are the effectiveness of tag-based and explanation-based interventions compare to conditions with no interventions in the short term and long term from readers' aspect?

2. Are model-generated explanations really more effective than warning tags in both time frames?

## 2 Related Works

### 2.1 Effectiveness of Misinformation Warning Labels

When a piece of fake news is detected by the system (Horne and Adali, 2017) or reported by online users (Keith, 2021; Henry, 2019), a warning label is attached to the fake news. Recent studies have examined various aspects of misinformation warning labels, including label wording (Clayton et al., 2020; Kaiser et al., 2021), warning icons (Kaiser et al., 2021; Moravec et al., 2020), and labels provided by fact-checkers (Pennycook et al., 2018), algorithms (Seo et al., 2019), or online community users (Jia et al., 2022; Yaqub et al., 2020). The results collectively show that specific warning labels (e.g., "rated false") are more effective than general warning labels (e.g., "disputed"). Also, misinformation warning labels attributed to third-party fact-checkers, algorithms, or online users can produce similar effects on reducing participants' beliefs.

Despite the significant progresses, most previous studies have focused on examining the effect of misinformation warning labels immediately (i.e., a short-term effect). The continued-influence affect (Anderson et al., 1980; Lewandowsky et al., 2012)—the persistence of misbelief in misinformation even after a warning — highlights the importance of evaluating *long-term* effect of misinformation warning labels. Few studies have addressed the long-term effect of warning labels, and results generally show that the effect of such labels decays over time (Pennycook et al., 2018; Seo et al., 2019; Grady et al., 2021). To mitigate

the persistence of misbelief in misinformation, Lo et al. (2022) proposed an implicit approach to deliver fake news verification such that fake-news readers can continuously access verified news articles about fake-news events without explicit correction. Compared to a fact-checker warning label condition, the results of such implicit intervention showed that participants maintained or slightly increased their sensitivity in differentiating real and fake news in the long term. Such results shed light on the importance of assisting users in distinguishing false information from factual contents for mitigating misinformation.

### 2.2 Increasing Focus on Fact-Checked Explanations

Researchers have adopted NLP techniques to provide explanations for detected misinformation, for instance by highlighting biased statements (Baly et al., 2018; Horne et al., 2019). Most current explanation-based methods frame the task of misinformation mitigation as text summarization. Specifically, such methods locate appropriate fact-checked evidence which is then summarized as an explanation (Atanasova et al., 2020; Kotonya and Toni, 2020). However, because the nature of summarization is to generate a paragraph that best represents the meaning of all documents of evidence, this could fail to explain *why* a piece of information is false.

Dai et al. (2022) proposed a framework to generate fact-checked counterfactual explanations (Byrne, 2016), the idea of which is to construct another instance showing minimal changes on evidence that result in a different fact-checking prediction by models (Mittelstadt et al., 2019). Dai et al. also conducted online human-subject experiments and obtained results showing that the counterfactual explanations outperformed summarization-based explanations. Such initial efforts yield a preliminary understanding of fact-checked counterfactual explanations in mitigating misinformation. To the best of our knowledge, the long-term effect of counterfactual explanations on mitigating humans' belief in misinformation has not been examined.

## 3 Experiment:
## Warning Tag or Explanation?

As current automated metrics frequently show weak correlation with human evaluations of

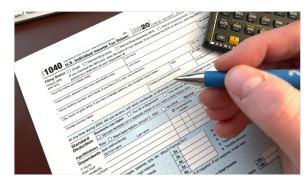
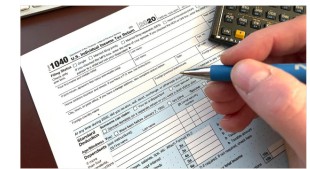
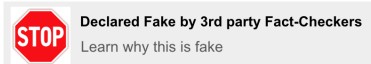
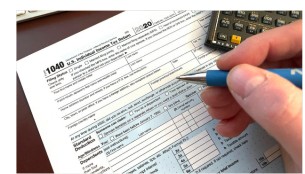
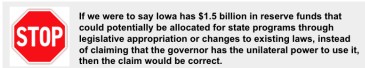

Figure 1: Example of a fake claim across the three conditions: the Control condition (left panel), the Warning-Tag (WT) condition (center panel), and the Counterfactual-Explanation (CF-E) condition (right panel).

explanation quality (Clinciu et al., 2021), the current study focuses on human evaluation. We conduct an online human-subject study using a mixed design. Specifically, participants evaluate real and fake claims across different time frames (pre-test, post-test, and long-term test) in three between-subject conditions [Control, Warning Tag (WT), and Counterfactual Explanation (CF-E)].

## 3.1 Materials

We sample 24 news claims (half fake and half real) from PolitiFact.com[4]. Each claim has a veracity label and evidence from professional fact-checkers. Half of the claims are pro-Democratic, and the other half are pro-Republican. We follow LIAR-PLUS (Alhindi et al., 2018) to collect the evidence for each news claim. The news claims are selected through a pilot study [5] to balance intensity of partisanship, familiarity level, and perceive accuracy between pro-Democratic and pro-Republican claims. Explanations were generated using GPT-4 (OpenAI, 2023), with a prompt based on the state-of-the-art counterfactual explanation form (Dai et al., 2022):

*Claim: <input>, Evidence: <input> This is a false claim. Please generate a short sentence of a counterfactual explanation to the claim. The sentence structure should be "If we were to say ... instead of ... the claim would be correct".*

We manually verify the accuracy of AI-generated explanations to ensure they align with evidence from human experts and are free from

errors. All news claims and generated CF-Es for fake news can be found in Tables 3 and 4.

A stop sign with warning texts are employed in the warning tag. Such an intervention method has been shown to be the most effective warning tag on social media (Moravec et al., 2020). We replace the warning texts with a model-generated counterfactual explanation. Figure 1 center and right panels show the intervention interfaces.

## 3.2 Procedure

After informed consent and a brief introduction of the study, participants are randomly assigned to one of the three conditions.

**Pre-test Phase (Baseline)** Participants in all conditions start with evaluating four fake claims and four real claims without any intervention. Participants indicate their familiarity with each claim using a five-point scale: *Have you ever seen or heard about this claim?* (1 = Definitely not, 5 = Definitely yes) and perceived accuracy using a seven-point scale (Sindermann et al., 2021): *To the best of your knowledge, how accurate is the claim?* (−3 = Definitely not accurate, 3 = Definitely accurate). Because confirmation bias has been found to influence users' beliefs in news claims (Ling, 2020), we also collect the participants' confirmation bias on each claim (Moravec et al., 2020) by asking their perceived importance using a seven-point scale: *Do you find the issue described in the article important?* (1 = Not important at all, 7 = Extremely important). Confirmation bias is measured by multiplying the perceived accuracy and importance ratings, creating a scale from −21 to 21. Statistic results are shown in Appendix A .

---

[4]https://www.politifact.com/ A website primarily features news from the United States.

[5]The pilot study was conducted with 72 annotators located in the United States. Those in the pilot did not participate in the main study, ensuring no overlap.

| | Accuracy Rate | | Flip Rate (Short-term/ Long-term) | | | | |
|---|---|---|---|---|---|---|---|
| | Pre-test | Post-test/ Long-term | ✗ → ✓ | ▲→✓ | ▲→✗ | ✓ → ✗ | Overall |
| **Fake Claims with intervention in reading environment** | | | | | | | |
| CF-E | 41% | **77% / 69%** | **20% / 17%** | **22% / 18%** | 5% / 4% | 3% /4% | 49% /43% |
| WT | 40% | **72% / 66%** | **17% / 17%** | **20% / 17%** | 1% / 3% | 2% / 3% | 40% / 39% |
| **Fake Claims without intervention in reading environment** | | | | | | | |
| Control | 40% | 38% / 38% | 4% / 7% | 8% / 8% | 7%/ 8% | 7%/ 10% | 26%/ 32% |
| **Real Claims without intervention in reading environment** | | | | | | | |
| CF-E | 35% | 44% / 40% | 8%/ 8% | 13%/ 7% | 13%/ 7% | 6%/ 6% | 33%/ 31% |
| WT | 33% | 52% / 49% | 13%/ 13% | 13%/ 12% | 7%/ 7% | 3%/ 6% | 36%/ 39% |
| Control | 35% | 37% / 34% | 7%/ 5% | 7%/ 7% | 4%/ 6% | 4%/ 4% | 22%/ 21% |

Table 1: Accuracy rate shows the proportion of participants' responses answered correctly to the accuracy of the news claims in the pre-test and post-test stages. The flip rate states the proportion of outcomes that were flipped from pre-test to post-test (short-term)and pre-test to long-term test (long-term ). ✗→✓ represents the proportion of participants' answers incorrectly in the pre-test but correctly in the post-test or long-term test. ✓→✗ represents the proportion of participants' answers correctly in the pre-test but incorrectly in the post-test or long-term test. ▲ represents the participants answer "Might or might not be accurate" in the pre-test.

**Intervention Phase** Participants in each condition read the same eight news claims as in the pre-test phase. While participants in the Control condition view the fake claims without any intervention, those in the intervention conditions read each fake claim with the corresponding warning tag or explanations. Figure 1 shows the interfaces of each condition.

**Questionnaire** We solicit participants' opinions on the fact-checked debunking strategies. We ask them about their perception of fake news debunking strategy using a five-point scale: *In the previous module, do you see any fake news debunking strategy?* (1 = Definitely not, 5 = Definitely yes). Other multi-choice questions are about the helpfulness and the overall experience of the survey. We administer the questionnaire in this stage to prevent the debunking strategies presented in the intervention phase from rehearsing in participants' working memory at the post-test evaluation.

**Post-test Phase** Participants re-evaluate the same news claim as the previous phases. We again ask about their perceived accuracy of the news claims to determine whether the fake news debunking strategies has flipped their initial evaluation of each claim. We further ask the reasons of the perceived accuracy rating by a multi-choice question: *Why did you choose the answer of the previous question?* (1: I chose this answer base on my previous knowledge; 2: because I saw the tag/ explanation; 3: because I searched online; 4: others).

**Long-term test Phase** Finally, we conduct a long-term test to determine whether the debunking strategies have a long-term effect on participants' perceived accuracy of news claims. A delay of one or two days has been used in psychology literature to access long-term memory performance (Frith et al., 2017; Chan et al., 2006; Nairne, 1992). Thus, 24 hours after the post-test, we invite all participants back to inquire about their perceptions of the news claims' accuracy, using a set of questions same as the post-test. We invite the unreturned participants again after a 36-hour wait.

### 3.3 Recruitment

We recruited participants (N = 270) on Prolific (Palan and Schitter, 2018). We required our participants to be located in the United States, which is critical to ensure they have cultural and contextual understanding of the selected news claims and the counterfactual explanations. For each question, we required a minimum reading and selection time of 5 seconds before a submission button shows up. We discarded annotations completed in an unreasonably short time, reported not encountering any fake news debunking strategy in the WT or CF-E groups, or without completing the long-term test. We included 215 participants' responses in the data analysis.

### 3.4 Results

Results are shown in Table 1. Our main findings are as follows:

**Fact-Checked intervention strategies indeed help humans debunk fake claims immediately:** As shown in Table 1, participants made more accurate veracity judgments for the fake claims after viewing the warning tags or explanations. The accuracy rate of fake news improved from 41% at pre-test phase to 77% at post-test phase for the CF-E group. The accuracy rate of the WT group showed a 32% increase (pre-test:40% → post-test: 72%). Yet, the accuracy rate of the fake claims for the Control group remained similar in both pre-test (40%) and post-test (38%). While chi-squared test suggests that interventions helped participants understand that a news claim is false ($\chi^2_{(2)} = 39.6, p < 0.001$), the improvements were statistically indistinguishable between the two intervention groups ($\chi^2_{(1)} < 1, p = 0.65$). Moreover, the flip rates of inaccurate or ambiguous ratings to accurate ratings are around 40% in both intervention groups, indicating that the intervention methods change participants' perception of the misinformation substantially in the short-term.

**Explanations are not more effective in the long-term:** Our hypothesis is that the explanation-based intervention would be more effective in debunking fake news than the tag-based intervention in the long-term as the explanations provides more information to the participants concerning the reason for refuting the fake news. While the long-term effects of the intervention groups were evident ($\chi^2_{(2)} = 32, p < 0.001$, the effects were similar between the CF-E (a 28% increase from the pre-test to the long-term test) and WT (a 26% increase from the pre-test to the long-term test) conditions ($\chi^2_{(1)} < 1, p = 0.99$). These results suggest that both intervention methods demonstrate short-term and long-term effectiveness.

We conjecture two possible reasons for the obtained null significant results in the long-term. On one hand, we placed a stop-sign warning in the explanation condition. Prior studies have shown that such design could make participants pay more attention to the icons but fail to notice the texts (Kaiser et al., 2021). The red color has been widely used in risk communication (Wogalter et al., 1999), which could also make participants react upon the color quickly without taking the time to read the explanations. One the other hand, the font size and the length of the explanations could have made participants less motivated to check

the warning details (Samuels, 1983; Kadayat and Eika, 2020). Future work should use comparative effectiveness studies to isolate the effects of the warning icon and the explanations. Additionally, we compared the short-term and long-term results of the two intervention conditions and found no significant differences ($\chi^2_{(1)} < 1, p = 0.99$). These results imply that a longer delay beyond 48 hours such as one week (Pennycook et al., 2018; Seo et al., 2019) could be considered in future work to further test the long-term effect.

**Potential Source Searching** We analyzed reasons for participants' choices during both the post- and long-term tests; the details of the question and options are described in Section 3.2. While more participants in the Control condition sought out additional online information than those in the two intervention conditions in the post-test, such pattern was reversed in the long-term test. In particular, participants in the intervention conditions showed a similar or slightly higher online searching rate but those in the Control condition reduced the search. Such results are interesting and imply that both interventions can potentially promote fact-checking and foster deeper engagement with news. See details in Appendix B and Table 5.

## 4 Conclusion

We demonstrate that both tag-based and explanation-based methods can effectively combat fake information in the short- and long-term. However, such findings suggest that an emphasis on generating improved explanations from the aspects of NLP is not sufficient to address the primary challenge in mitigating humans' belief in misinformation. To make the NLP-based debunking strategies more effective in the long-term and further reduce recurred misbeliefs, it is crucial to consider visual design choices to boost readers' motivation to engage with detailed yet accessible explanations, such as font size, text length, or even personalized explanations.

## Limitations

Our study assesses tag-based and explanation-based strategies independently, not jointly. This decision was made to isolate the effects of each method and avoid possible confounding influences.

However, warning labels and machine-generated explanations could be used in conjunction to augment the effectiveness of misinformation debunking in a real-world setting. This potential for synergy between the methods could provide a more robust approach to combat fake news; future research could benefit from exploring this joint application.

Indeed, another limitation of our research lies in its examination of tag-based debunking strategies. Various types of warning labels are employed in different contexts, each with its own design, wording, and positioning. These variables may significantly influence the effectiveness of a warning label. In our study, we examine a specific type of warning label, and although our results provide important insights, they may not be representative of the potential effectiveness of all types of warning labels.

Also, in our case, we only sampled explanations from those that had been examined as having no system error. However, the generation model might have errors and thus generate low-quality fact-checked explanations.

Finally, in real-life situations, the fake news detector may mistakenly label genuine news as fake, which wasn't a factor we considered in our study. Instead, we relied on the expertise of fact-checkers from ProliticFact.com to provide labels. However, if the detector were to flag real news as fake in practice, it could lead to issues associated with the warning labels.

## Ethics Statement

Our study has been approved by the Institutional Review Board of the authors' institution. We obtained informed consent from each participant. We acknowledge that participants were inherently exposed to the risk of reading fake news. However, prior studies showed that misinformation studies did not significantly increase participants' long-term susceptibility to misinformation used in the experiments (Murphy et al., 2020). As our survey is a two-wave study which did not force participants to finish the whole study, participants could receive partial payment right after completing the short-term study. We also informed participants that they had the right to withdraw from the experiment whenever they desired. Participants were paid based on a rate of $8/ hour, which is above the federal minimum wage in the United States.

## Acknowledgements

This research is supported by National Science and Technology Council of Taiwan, under Grants no. 111-2221-E-001-021- and 111-2634-F-002-022-. The works of Aiping Xiong were in part supported by NSF awards #1915801 and #2121097.

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

## A  Demographic of Participants

| Item | Options | *Survey* | N |
|------|---------|----------|---|
| | Female | 37.02% | 82 |
| Sex | Male | 62.9% | 131 |
| | Other | 0% | |
| | 18–24 | 7.63% | 14 |
| | 25–34 | 29.77% | 63 |
| Age | 35–44 | 19.46% | 46 |
| | 45–54 | 18.70% | 38 |
| | Over 55 | 23.28% | 52 |
| | Black | 16.6% | 36 |
| | White | 69.9% | 151 |
| Ethnicity | Asian | 5.55% | 12 |
| | More than one race | 5.09% | 11 |
| | Other | 2.77% | 6 |

Table 2: Demographic information of valid participants in our survey.

The ratio of familiar claims are 0.25 for fake news ($\chi^2_{(2)} < 1$, $p = 0.994$) and 0.2 for real news ($\chi^2_{(2)} < 1$, $p = 0.991$) respectively with no significant difference across all condition groups. The ratio of perceived important claims are 0.59 for fake news ($\chi^2_{(2)} < 1$, $p = 0.999$) and 0.51 for real news ($\chi^2_{(2)} < 1$, $p = 0.999$) with no significant difference across all condition groups respectively either. The average confirmation bias are -1.31 ($\chi^2_{(2)} < 1$, $p = 0.956$) for fake news and -1.46 ($\chi^2_{(2)} < 1$, $p = 0.944$) for real news with no significant difference across all condition groups.

## B  Participants' Reason of Choice

For fake claims with interventions in the reading environment, the CF-E and WT groups show different distributions in their reliance on previous knowledge and online searching, as depicted in Table 5. The control group heavily depends on online searching, particularly with real claims, similar to CF-E and WT groups for real claims. Only a negligible proportion of participants reported searching online for extra information during both short-term and long-term tests in both intervention groups. We consider this as a positive impact if participants searched online for external information, as our intervention's goal isn't just misinformation correction but also behavioral change—specifically, fact-checking consumed news. Online searching indicates that the interventions motivate deeper news engagement by encouraging further fact-seeking.

| | Fake News Claim | Model-generated Counterfactual Explanation |
|---|---|---|
| 1 | Sen. Ron Johnson again says Social Security is a Ponzi scheme | If we were to say Social Security operates similarly to a Ponzi scheme in terms of financial structure but with key differences that ensure benefit payments and government obligations, the claim would be correct. |
| 2 | The average tax for billionaires is about 3%, which is a lower tax than a schoolteacher or a firefighter. | If we were to say the average tax for billionaires is about 16%, which is higher than the effective tax rates of many teachers and firefighters falling within the $50,000 to $100,000 income range, the claim would be correct. |
| 3 | The largest contributor to the debt ceiling, or to our deficit, has been the Trump tax cuts. | If we were to say the Trump tax cuts have contributed to the debt ceiling and deficit, but other bills such as the 2019 spending bill, the CARES Act, and the American Rescue Plan, which received Democratic support, have added nearly as much or more to the debt, the claim would be correct. |
| 4 | $1.5 billion of your taxpayer money is sitting in an account that can be used for other purposes. | If we were to say Iowa has $1.5 billion in reserve funds that could potentially be allocated for state programs through legislative appropriation or changes to existing laws, instead of claiming that the governor has the unilateral power to use it, then the claim would be correct. |
| 5 | New York City is wasting taxpayer money on billboards in Florida. | If we were to say 'New York City is benefiting from pro bono creative content and donated billboard space in Florida' instead of 'New York City is wasting taxpayer money on billboards in Florida', the claim would be correct. |
| 6 | The average federal income tax paid by the richest Americans is 8%. ... If you're a cop, a teacher, a firefighter, union worker, you probably pay two to three times that. | If we were to say the average federal income tax paid by the richest Americans is 20% of their income, while many households earning between $50,000 and $100,000 a year pay effective tax rates between 0% and 15%, the claim would be correct, as it reflects the actual tax rates under the current tax code. |
| 7 | Both murderers in Texas were not only illegals, but gang members and were most likely tied to MS-13. | If we were to say 'Both murderers in Texas were U.S. citizens, with one having an extensive criminal history, but no confirmed gang affiliations' instead of 'Both murderers in Texas were not only illegals, but gang members and most likely tied to MS-13,' the claim would be correct. |
| 8 | Only about 20% of people who apply for asylum qualify for it. | If we were to say that only about 30% of people who apply for asylum qualify for it, based on the approval rate since 2000 according to data from Syracuse University, instead of 20%, the claim would be correct. |
| 9 | As president, Donald Trump "marshaled the full power of government to stop deadly drugs, opioids, and fentanyl from coming into our country. As a result, drug overdose deaths declined nationwide for the first time in nearly 30 years. | If we were to say that Donald Trump's efforts to combat drug overdoses were effective in reducing overdose deaths from 2017 to 2018, but failed to sustain progress in the following years, including a record increase in 2020, the claim would be correct. |
| 10 | Says J.D. Vance said, "People who voted for Trump voted for him for racist reasons. | If we were to say 'J.D. Vance acknowledged that some people may have voted for Trump for racist reasons, but argued that most people voted for his policies on jobs' instead of 'People who voted for Trump voted for him for racist reasons,' the claim would be correct. |
| 11 | Biden and Democrats have dismantled border security. | If we were to say the Biden administration has maintained comparable budgets for border security, enforced border laws and policies, and utilized funds to repair barriers and close gaps, but halted the construction of additional miles of barriers, the claim would be correct. |
| 12 | Biden said there was a "28% increase in children to the border in my administration" and "31% ... in 2019." The increase in migration in January, February and March "happens every year. | If we were to say there was a 63% increase in children arriving at the border during the Biden administration, which is more than twice the percentage he mentioned, and that the total number of encounters this year is likely to be the highest in 20 years, instead of focusing solely on the seasonal pattern of increased migration, the claim would be correct. |

Table 3: All fake news claims and the corresponding model-generated counterfactual explanations

| Real News Claim | |
|---|---|
| 1 | We're the only major gas-producing state in the US that doesn't have a severance tax. |
| 2 | Florida has the second lowest tax burden per capita in the United States. |
| 3 | Since a new immigration program was implemented, the number of Venezuelans trying to enter the U.S. illegally decreased "from about 1,100 per day to less than 250 per day on average. |
| 4 | More fentanyl has crossed the border in the last two months under Biden than in 2019 under Trump. |
| 5 | WV families making less than $400K small businesses will NOT be targeted by the IRS. |
| 6 | The tax carve out (Ron) Johnson spearheaded overwhelmingly benefited the wealthiest, over small businesses. |
| 7 | Last year the IRS audited Americans earning less than $25,000 a year at five times the rate of other groups. |
| 8 | Virginia tax receipts in just the last four years alone have grown 50%. |
| 9 | Title 42 and other Trump-era holdovers are forcing migrants into dangerous, overcrowded conditions in Mexico. |
| 10 | Just last year," Miami-Dade Public Schools "had over 14,000 new children, 10,000 of which came from four countries of Cuba, Nicaragua, Venezuela and Haiti. |
| 11 | Approximately 60,000 Canadians currently live undocumented in the USA. |
| 12 | Twice as many children are in Border Patrol custody under Biden than Trump peak in 2019. |

Table 4: All real news claims

| Reason of Choice (Short-term/ Long-term) | | | | |
|---|---|---|---|---|
| | Intervention | Previous Knowledge | Online Searching | Others |
| **Fake Claims with intervention in reading environment** | | | | |
| CF-E | 66.5%/ 42.7% | 30.4%/ 52.7% | 1.5%/ 1.9% | 1.5%/ 1.2% |
| WT | 58.8%/ 37.7% | 36.4%/ 59.1% | 1.9%/ 1.6% | 2.8%/ 1.6% |
| **Fake Claims without intervention in reading environment** | | | | |
| Control | 12.9%/ 9.8% | 80.5%/ 87.5% | 3.1%/ 0.8% | 3.5%/ 0.4% |
| **Real Claims without intervention in reading environment** | | | | |
| CF-E | 19.6%/ 14.6% | 72.7%/ 80.4% | 1.9%/ 1.2% | 5.8%/ 2.3% |
| WT | 32.5%/ 19.8% | 60.4%/ 73.4% | 1.9%/ 2.6% | 4.5%/ 4.2% |
| Control | 12.9%/ 9.8% | 80.5%/ 86.3% | 3.1%/ 0.8% | 3.5%/ 1.6% |

Table 5: The table presents a breakdown of participants' reasons for their choices in the short and long term when exposed to fake and real news claims under different intervention methods. The reasons include relying on previous knowledge, online searching, and other factors.