# OpenReview forum: "Is Explanation the Cure? Misinformation Mitigation in the Short Term and Long Term"
_EMNLP/2023/Conference — EMNLP 2023 Findings_

### Official Review · Reviewer_1tRC · 2023-08-02

**Soundness:** 2

**Excitement:**

2: Mediocre: This paper makes marginal contributions (vs non-contemporaneous work), so I would rather not see it in the conference.

**Paper Topic And Main Contributions:**

The paper compares the effectiveness of fake news warning labels and the counterfactual explanations generated by GPT-4 as debunking
methods with a human study. According to a study involving 215 participants and 24 news claims, the authors assert that interventions aimed at debunking help readers recognize the falseness of news claims in both the short-term and long-term, although to a somewhat lesser degree in the long-term.

**Questions For The Authors:**

1. have you tried to vary the style/length of the generated explanations to see their influence?

2. how do you control the quality of the human study?

**Reasons To Accept:**

1. The research question is interesting and has practical applications in real life.

2. The paper proposes a human study to compare the two debunking options.

**Reasons To Reject:**

1. The paper lacks results and analysis. The only results included in the paper are from participants’ responses in Table 1. It would be much better to include some automatic evaluation metrics to justify the findings, or the results would be subjective and weak.

2. There are several factors that could affect the experiments, but details are missing. For example, the quality of the generated explanations, detailed prompts to use, length/style of the explanations, etc.

3. I don't see how the authors justify the quality of the human study. Indicators like inner agreement should be included to make the study more convincing.

4. The information volume within the explanations and the warning tag are very different, so it is not even a fair comparison. e.g. people may not be willing to read the whole lengthy explanation.

5. I have read the rebuttal and don't think the human study here is designed carefully and thoroughly. Besides, this seems to fit more for a psychology conference instead of an NLP conference. Actually, the paper is nothing related to NLP except using GPT-4 for some explanation generation.

**Reproducibility:**

1: Could not reproduce the results here no matter how hard they tried.

**Reviewer Confidence:**

3: Pretty sure, but there's a chance I missed something. Although I have a good feel for this area in general, I did not carefully check the paper's details, e.g., the math, experimental design, or novelty.

---

> ### Author Rebuttal · Authors · 2023-08-27
>
> We thank Reviewer 1tRC for recognizing the practicalness and showing interest in our research. Your feedback is deeply appreciated. The following are our responses to the reject reasons of Reviewer 1tRC:
>
> **R1: Reason of Not Including Automatic Evaluations**
>
> We would like to highlight that the primary objective of our study is not to assess the generative capabilities of the model, especially since GPT-4 has already established its performance in most of NLP tasks [2]. We aim to evaluate the effectiveness of model generated explanations and warning tags. As we mentioned in the Lines 190-192 of our manuscript, previous study has suggested that existing automatic metrics correlate poorly with human judgments of quality[1, 4, 5]. Given this, we believe that relying on human’s responses provides a more accurate measure of the effectiveness of explanation and warning tag interventions.
>
> **R2, Q1: Details of Experiment**
>
> **Quality of the Generated Explanations:** GPT-4 has already established its performance in most of NLP tasks [2]. Furthermore, we manually reviewed the explanations generated by the model, excluding any inaccuracies, ensuring our experiment's results are not affected by potential GPT-4 model errors. Examples of model generated explanations can be referred to Table 3 in the paper.
>
> **Detailed Prompt:** We include the GPT-4 prompt for explanation generation in Lines 229-233.
>
> **Length and Style of the Explanations:** We utilize single-sentence counterfactual explanations, proven to be more effective than both abstractive and extractive summary-based explanations for debunking misinformation, as elaborated in Lines 78-81 of our manuscript and further explored in [3]. Rather than restricting sentence length, we confined the structure to counterfactual explanations, giving GPT-4 the flexibility to generate thorough and coherent sentences, which on average consist of 45 words. This supplemental detail will be included in our revised paper for enhanced clarity.
>
> **R3: Reason of Not Including Inner Agreements**
>
> Our study's core focus is on exploring individual differences in opinions, not on achieving uniform labels or judgments on a given task. Also, inner agreements are mainly used for qualitative responses, which is not the focus of our study. Therefore, inner agreements are not applicable or relevant for evaluating our study's quality. We'll clarify this point in the revised manuscript to address any concerns.
>
> **R4: Difference of Information Volume in Warning Tags and Explanations**
>
> In fact, highlighting the potential unwillingness of individuals to engage with explanations which include higher information volume is one of the issues our study aims to raise. By comparing more warning tags with longer, more detailed explanations, our study seeks to compare the effectiveness in the context of combating misinformation. The observed results, which indicate no significant difference in effectiveness between the warning tags and explanations, provide valuable insights for designing debunking misinformation strategies. We will clarify this aspect of our research focus in the revised manuscript to make our intentions and findings more transparent.
>
> **Q2: Quality Control of Human Study**
>
> To ensure the quality of our human study, we implemented several criteria as outlined in the paper (Lines 194-198). If an annotator completed the task in an unreasonably short timeframe, indicated they did not come across any fake news debunking strategy in the warning tag or explanation groups, or failed to complete the long-term test, we excluded their annotations. Additionally, any annotations from participants who reported encountering a fake news debunking strategy within the control group are also discarded. These measures helped us maintain a high standard of data integrity and consistency throughout the study.
>
> **References**
>
> [1] Miruna-Adriana Clinciu, Arash Eshghi, Helen Hastie, A Study of Automatic Metrics for the Evaluation of Natural Language  Explanations EACL 2021, https://aclanthology.org/2021.eacl-main.202
>
> [2] OpenAI, GPT-4 Technical Report, 2023, https://arxiv.org/abs/2303.08774
>
> [3] Shih-Chieh Dai, Yi-Li Hsu, Aiping Xiong, and Lun-Wei Ku. Ask to Know More: Generating Counterfactual Explanations for Fake Claims. KDD '22 https://dl.acm.org/doi/abs/10.1145/3534678.3539205
>
> [4] Maxime Peyrard, Studying Summarization Evaluation Metrics in the Appropriate Scoring Range , ACL 2019, https://aclanthology.org/P19-1502
>
> [5] Daniel Deutsch, Rotem Dror, and Dan Roth, Re-Examining System-Level Correlations of Automatic Summarization Evaluation Metrics, NAACL 2021, https://aclanthology.org/2022.naacl-main.442.pdf

---

### Official Review · Reviewer_fdhr · 2023-08-02

**Soundness:** 3

**Excitement:**

4: Strong: This paper deepens the understanding of some phenomenon or lowers the barriers to an existing research direction.

**Paper Topic And Main Contributions:**

This paper presents a study investigating the effectiveness of two approaches to debunking fake news on social media: warning tags and AI-generated explanations. The study was conducted in two waves, involving participants who were divided into a control group and two treatment groups. The control group was exposed to false content without any intervention, while one treatment group received false content with warning tags, and the other had false news accompanied by explanations generated by the GPT-4 model.

The study found that both the warning tags and the AI-generated explanations significantly decreased immediate belief in fake news, with no notable difference between the two approaches. The two methods also had long-term impacts on participants' self-reported belief in fake news. The authors suggest that future research could explore the combined use of warning tags and AI-generated explanations and take into consideration various designs of warning tags.

**Questions For The Authors:**

1. Did you check whether your subjects looked for the sources of news between the two tests? If so, how would that affect the results?
2. How did you evaluate the GPT-4 explanations generated and presented to the participants?
3. How did you choose the news for evaluation? Were they related to your subjects' demographic?
4, Did you have any precautions to prevent the subjects from labeling the news from memory?

**Reasons To Accept:**

1. This study addresses the pressing issue of fake news on social media, which has significant real-world implications. Given the pervasiveness of misinformation, the effectiveness of various debunking approaches is a highly relevant area of research.
2. This work brings a novelty by focusing on the short-term and long-term effects of debunking fake news. This adds a dimension of depth to the study, allowing for a more comprehensive understanding of the effects of debunking techniques.
3. This research raises an important question regarding the effect of intervention methods. This fosters ongoing discourse and encourages further exploration in this field.
4. While the study needs more analysis, it provides empirical evidence that warning tags and AI-generated explanations can significantly lower the immediate belief in fake news. This is important as it offers tangible solutions that social media platforms can consider implementing.

**Reasons To Reject:**

1.  While the study's use of AI-generated explanations is innovative, there is a potential challenge when the generated counterfactuals may not be completely accurate. This could further propagate misunderstandings or misinformation.
2. The paper could provide additional detail on the nature of the news items that were annotated by the subjects. Greater transparency in this area could help in understanding the full context and applicability of the findings.
3. There seems to be a discrepancy between the described experimental setup and the interface depicted in Figure 3. Clarification or harmonization will help avoid potential confusion for readers.
4. The hypothesis put forth in Line 278 could be better substantiated, either through explicit instructions to the subjects or through additional surveying. Addressing this point more thoroughly could further strengthen the evidence supporting the authors' claims.
5. While the research is commendable for bringing attention to the effectiveness of different intervention methods, a more in-depth analysis of the subjects' decision-making process could further enhance the robustness of the supportive argument. This aspect is worth exploring more comprehensively to clarify the complex dynamics at play further.

**Reproducibility:**

3: Could reproduce the results with some difficulty. The settings of parameters are underspecified or subjectively determined; the training/evaluation data are not widely available.

**Reviewer Confidence:**

3: Pretty sure, but there's a chance I missed something. Although I have a good feel for this area in general, I did not carefully check the paper's details, e.g., the math, experimental design, or novelty.

---

> ### Author Rebuttal · Authors · 2023-08-27
>
> We thank Reviewer fdhr for recognizing the significance and depth of our research on debunking fake news, as well as its potential to foster further exploration and offer tangible solutions to social media platforms. Your feedback is deeply appreciated. The following are our responses to the reject reasons of Reviewer fdhr:
>
> **R1, Q2: Quality of Generated Explanations**
>
> Thank you for pointing out the potential challenge of AI-generated explanations. We'd like to clarify that our study exclusively uses AI-generated explanations that have been manually verified for accuracy by us. This verification process is based on checking the explanations are aligned with the evidence written by human experts from PolitiFact.com, ensuring not only factual accuracy but also grammatical integrity and logical coherence. While we acknowledge the concerns surrounding the accuracy of AI outputs, it is not the primary focus of our paper. We made deliberate efforts to ensure the reliability and accuracy of the data presented.
>
> **R2, Q3: Details of Selected Claims**
>
> We would like to highlight that the nature of the news items annotated by the subjects is included in Appendix 1. We followed the LIAR-PLUS dataset methodology to collect evidence for the 24 news claims from PolitiFact.com. To avoid extreme political leanings, familiarity, and perceived accuracy, the news claims were carefully selected through a pilot study conducted by additional 72 U.S. Prolific workers. Specifically, our study focuses on issues related to tax and immigrants issues in the US. We appreciate your suggestion and will ensure to include this information prominently in the main content of the final version. Additionally, the full list of the 24 news claims will be made available in the appendix. We hope these changes will address your concerns and enhance the clarity and transparency of our work.
>
> **R3: Experimental Setup Description**
>
> Thank you for pointing out the potential discrepancy between the described experimental setup and Figure 3. We would like to clarify that Figure 3 showcases one of the samples presented to the subjects in our experiment. We will carefully review and adjust the description of the experimental setup to reduce any confusion.
>
> **R4: Influence of Font Size and Sentence Length**
>
> Thank you for your thoughtful feedback on Line 278. We agree that our statement could benefit from more robust substantiation. Existing studies have highlighted the influence of sentence length on both comprehension and workload, particularly in the context of screen readers [2]. Additional research also points out that text size and sentence length can affect reading speed, comprehension, and even motivation to read [3]. This aligns with our hypothesis in Line 278 that readability factors—like sentence length and font size—could influence the effectiveness of explanations. We will include this information in the revised manuscript for a more comprehensive discussion.
>
> **Q1: Potential Source Exploration**
>
> We appreciate your advice and we will include the following additional information in the revised paper. We specifically asked participants why they chose their respective answers to understand their rationale better. The question posed was, "Why do you choose the answer to the previous question?" and the options were based on previous knowledge, intervention effect (tag/explanation), online search, or others, as specified in Lines 631-636 of our manuscript.The below tables present the results of the question.
>
> Our data indicate that only a negligible proportion of participants reported independently searching online for extra information during both short-term and long-term tests across all groups, with percentages ranging from 0.8% to 3.1%. Importantly, we would consider this a positive impact of the intervention if participants searched online for external information. The reason is that our goal of intervention is not just misinformation correction but also behavioral change—specifically, checking the facts of consumed news information. Therefore, increased online searching could be seen as evidence that our interventions are motivating participants to engage more deeply with the news by seeking out additional facts.
>
> **Reason of Choice (Fake news)**
>
> |  | Intervention  (short-term/ long-term) | Previous Knowledge (short-term/ long-term) | Search Online (short-term/ long-term) | Others (short-term/ long-term) |
> | --- | --- | --- | --- | --- |
> | WT | 58.8%/ 37.7% | 36.4%/ 59.1% | 1.9%/ 1.6% | 2.8%/ 1.6% |
> | CF-E | 66.5%/ 42.7% | 30.4%/ 52.7% | 1.5%/ 1.9% | 1.5%/ 1.2% |
> | Control | 12.9%/ 9.8% | 80.5%/ 87.5% | 3.1%/ 0.8% | 3.5%/ 0.4% |
>
> **Reason of Choice (Real news)**
>
> |  | Intervention (short-term/ long-term) | Previous Knowledge (short-term/ long-term) | Search Online (short-term/ long-term) | Others (short-term/ long-term) |
> | --- | --- | --- | --- | --- |
> | WT | 32.5%/ 19.8% | 60.4%/ 73.4% | 1.9%/ 2.6% | 4.5%/ 4.2% |
> | CF-E | 19.6%/ 14.6% | 72.7%/ 80.4% | 1.9%/ 1.2% | 5.8%/ 2.3% |
> | Control | 12.9%/ 9.8% | 80.5%/ 86.3% | 3.1%/ 0.8% | 3.5%/ 1.6% |
>
> **Q4: Relying on Memory to Label**
>
> We did not take specific measures to prevent subjects from relying on memory when labeling news claims. However, the self-reported familiarity (Experimental details are in Line 585) with the claims is relatively low. The proportions of participants who answered "Definitely yes" or "Probably yes" were 0.25 and 0.2 for fake and real news, respectively. Statistical tests showed no significant difference in this respect across all groups, including the CF-E, WT, and control groups (χ^2(2) < 1, p = 0.994 for fake news; χ^2(2) < 1, p = 0.991 for real news). Moreover, previous research [1] has indicated that familiarity with claims does not significantly impact the effectiveness of explanations. Therefore, we believe that the influence of memory on our study's results is likely minimal.
>
> **References**
>
> [1] Shih-Chieh Dai, Yi-Li Hsu, Aiping Xiong, and Lun-Wei Ku. 2022. Ask to Know More: Generating Counterfactual Explanations for Fake Claims. KDD '22 https://dl.acm.org/doi/abs/10.1145/3534678.3539205
>
> [2] Kadayat, Bam Bahadur, and Evelyn Eika. Impact of Sentence Length on the Readability of Web for Screen Reader Users HCII’20 https://link.springer.com/chapter/10.1007/978-3-030-49282-3_18
>
> [3] S. Jav Samuels, A Cognitive Approach to Factors Influencing Reading Comprehension, The Journal of Educational Research 1983 https://www.jstor.org/stable/27539984

---

### Official Review · Reviewer_whUc · 2023-08-05

**Soundness:** 4

**Excitement:**

3: Ambivalent: It has merits (e.g., it reports state-of-the-art results, the idea is nice), but there are key weaknesses (e.g., it describes incremental work), and it can significantly benefit from another round of revision. However, I won't object to accepting it if my co-reviewers champion it.

**Paper Topic And Main Contributions:**

This research aims to combat fake news by addressing the effectiveness of debunking strategies with the assistance of NLP methods. In particular, the study focuses on tag- and counterfactual-based explanations and proposes a comprehensive evaluation of them over short and long-term periods. The research employs the GPT-4 model, a state-of-the-art LLM, to generate counterfactual explanations. Based on a human experiment involving 24 news claims and 215 participants, the authors claim that those debunking interventions help readers understand that the news claims are false in both the short-term and long-term, but to a lesser extent in the long-term.

**Questions For The Authors:**

N/A

**Reasons To Accept:**

A1. The issue of debunking fake news is of utmost importance for society, and this study aims to measure the effectiveness of various debunking strategies, which could bring significant implications.

A2. Notably, some findings, such as the effectiveness of counterfactual explanations generated by the GPT-4 model in reducing belief in fake information, are noteworthy and deserve attention.

**Reasons To Reject:**

R1. While it is a promising attempt to use LLMs for generating counterfactual explanations for fake news, it appears that the correctness of these explanations has not been adequately evaluated. Since LLMs are known to have hallucination effects, there is uncertainty about the reliability of the generated explanations.

R2. In spite of the limited space in a short paper, more details about the human experiment should be included in the main text. For instance, the number of news claims used for the experiments and other relevant information should be provided.

R3. It seems that the experiments use the same set of news claims for pre-, intervention, post-, and long-term tests for each group. Thus, there's a possibility that the results are an outcome of participants’ exposure to the correct answer in the intervention. And that makes the validity of measuring the effectiveness of debunking strategies doubtful. I wonder whether there's a more subtle way to measure such behavioral change.

R4. While the study is important, its contribution to the NLP community is unclear.

**Reproducibility:**

3: Could reproduce the results with some difficulty. The settings of parameters are underspecified or subjectively determined; the training/evaluation data are not widely available.

**Reviewer Confidence:**

4: Quite sure. I tried to check the important points carefully. It's unlikely, though conceivable, that I missed something that should affect my ratings.

---

> ### Author Rebuttal · Authors · 2023-08-27
>
> We thank Reviewer whUc for emphasizing the societal relevance and implications of our research on debunking fake news and for noting the value of our findings regarding the use of GPT-4 generated counterfactual explanations. The following are our responses to the reject reasons of Reviewer whUc:
>
> **R1: Hallucination Effects of LLMs**
>
> We acknowledged the potential for hallucination effects in LLMs. However, it's important to note that the primary objective of our study is not to assess the generative capabilities of the model, especially since GPT-4 has already established its performance in most of NLP tasks [1]. We aim to evaluate the effectiveness of model generated explanations and warning tags. To address concerns about model-generated inaccuracies, we'd like to clarify that we manually reviewed the explanations generated by the model, excluding any inaccuracies, ensuring our experiment's results are not affected by potential GPT-4 model errors. We apologize for the oversight in not detailing this process earlier in the submission and will include this clarification in the final manuscript.
>
> **R2: Human-subject Experiment**
>
> We appreciate your feedback regarding the inclusion of more details about the human-subject experiment. We detailed these aspects in the Appendix because of the page limitation. In light of your feedback, we will move this information to the main body of the paper in the final version because we will be given one additional page in the proceedings. We will ensure that an additional content page in the proceedings provides a comprehensive overview of the human experiment, including the number of news claims used and other relevant information.
>
> **R3: Exposure of Correct Answers**
>
> Thank you for highlighting the concern regarding the repeated use of the same set of news claims across various testing phases. However, our primary objective is to assess the effectiveness of different intervention methods, namely the explanation group, warning tag group, and control group. The structure of our study, which provides the correct answer during the intervention, is deliberately designed to measure shifts in individual stances or beliefs subsequent to that intervention.
>
> **R4: Contribution to the NLP Community**
>
> We believe our study bridges the gap between the NLP community and the societal challenge of fake news. Our research seeks to highlight the potential misdirection in the current NLP focus. The results suggest that even the state-of-the-art GPT-4 model’s explanations may not win over warning tags and can not sufficiently address the misinformation challenge. While there's value in enhancing explanation quality, our findings indicate that the primary challenge in combating misinformation might lie elsewhere. In our conclusions, we highlighted the need to raise users' motivation to engage with comprehensive yet digestible explanations. For NLP researchers, this might involve personalized explanations driven by Human-AI collaborative conversational systems, as mentioned in [2] and [3]. Such systems could dynamically adjust explanations to individual user preferences, cognitive styles, or prior knowledge, thereby making the information more relatable and actionable for the end-user. In the revised paper, we'll emphasize this perspective in the final version further to provide greater contribution to the NLP community.
>
> We hope these responses address the reviewer's concerns.
>
> **References**
>
> [1] OpenAI, GPT-4 Technical Report, 2023, https://arxiv.org/abs/2303.08774
>
> [2] Elliot Schumacher, Geoffrey Tso, Anitha Kannan, DERA: Enhancing Large Language Model Completions with Dialog-Enabled Resolving Agents, Varun Nair, 2023,  https://arxiv.org/abs/2303.17071
>
> [3] Monika Westphal, Michael Vössing, Gerhard Satzger, Galit B. Yom-Tov, Anat Rafaeli, Decision control and explanations in human-AI collaboration: Improving user perceptions and compliance,
> Computers in Human Behavior, 2023, https://doi.org/10.1016/j.chb.2023.107714.

---

### Meta-Review · Area_Chair_JBQv · 2023-09-14

**Recommendation:** 3

**Metareview:**

Following discussions with authors and among themselves, the reviewers overall found the paper sound.

The most prominent critiques with a consensus reached have to do with the lack of evaluation of the generated explanations, limited analyses of the main results, limited robustness tests, further necessary clarifications around the experimental design, and a narrow outline of contributions to the field.

---

### Decision · Program_Chairs · 2023-10-07

**Decision:**

Accept-Findings

**Comment:**

Following discussions with authors and among themselves, the reviewers overall found the paper sound.

The most prominent critiques with a consensus reached have to do with the lack of evaluation of the generated explanations, limited analyses of the main results, limited robustness tests, further necessary clarifications around the experimental design, and a narrow outline of contributions to the field.